# Risk factors for melioidosis in Udupi District, Karnataka, India, January 2017-July 2018

**Akhileshwar Singh[1], Ashok Talyan[1], Ramesh Chandra[1], Anubhav Srivastav[1], Vasudeva Upadhya[2], Chiranjay Mukhopadhyay[3,4], Shyamsundar Shreedhar[3], Deepak Sudhakaran[3], Suma Nair[3], Mohan Papanna[5,6], Rajesh Yadav[5]\*, Sujeet Kumar Singh[1], Tanzin Dikid[1]\***

**1** Epidemic Intelligence Service Programme, National Centre for Disease Control, Delhi, India, **2** District Surveillance Office, Udupi, Karnataka, India, **3** Kasturba Medical College, Manipal Academy of Higher Education, Manipal, Karnataka, India, **4** Center for Emerging and Tropical Diseases, Manipal Academy of Higher Education, Manipal, Karnataka, India, **5** Division of Global Health Protection, US Centers for Disease Control and Prevention, Atlanta, United States of America, **6** Huck Institute of Life Sciences, The Pennsylvania State University, PA, United States of America

☯ These authors contributed equally to this work.

\* mdx5@cdc.gov (RY); tanzindikid@gmail.com (TD)

**Data Availability Statement:** The database linked to this paper is available at: Dikid, Tanzin (2022), Meliodosis risk factors, Udupi, India, 2018, Dryad, Dataset, https://doi.org/10.5061/dryad.zcrjdfngh.

## Abstract

We initiated an epidemiological investigation following the death of a previously healthy 17 year-old boy with neuro-melioidosis. A case was defined as a culture-confirmed melioidosis patient from Udupi district admitted to hospital A from January 2013—July 2018. For the case control study, we enrolled a subset of cases admitted to hospital A from January 2017-July 2018. A control was resident of Udupi district admitted to hospital A in July 2018 with a non-infectious condition. Using a matched case-control design, we compared each case to 3 controls using age and sex groups. We assessed for risk factors related to water storage, activities of daily living, injuries and environmental exposures (three months prior to hospitalization), using conditional regression analysis. We identified 50 cases with case fatality rate 16%. Uncontrolled diabetes mellitus was present in 84% cases and 66% of cases occurred between May and October (rainy season). Percutaneous inoculation through exposure to stagnant water and injury leading to breakage in the skin were identified as an important mode of transmission. We used these findings to develop a surveillance case definition and initiated training of the district laboratory for melioidosis diagnosis.

## Introduction

Melioidosis is caused by gram-negative intracellular bacteria *Burkholderia pseudomallei*, which can infect both humans and animals. This environmental saprophyte is widely distributed in soil and fresh surface water in endemic regions of South East Asia, Northern Australia, the Indian subcontinent and areas of South America [1–4]. Up to 20% of community-acquired sepsis in the tropics is due to melioidosis and the overall mortality varies from 20–50% depending on the availability of healthcare services [5–8]. In 2015, the global burden of melioidosis was 4.6 million DALYs, which was higher than other common neglected tropical diseases [9]. Estimates suggest that the extent of melioidosis global distribution is widespread, and the

**Funding:** This public health activity was conducted by the India Epidemic Intelligence Service (EIS) officer as a response team deployed by the National Centre for Disease Control, New Delhi, India. The National Centre for Disease Control receives funding support for the India EIS Program through cooperative agreement No. NU2GGH001904GH10-1001 from the U.S. Centers for Disease Control and Prevention, Center for Global Health, Division of Global Health Protection. The funders had no role in study design, data collection, analysis, decision to publish, or manuscript preparation.

**Competing interests:** The authors have declared that no competing interests exist.

cases are grossly under-reported in 45 countries currently reporting [10]. This disparity is partly due to under-recognition due to its diverse clinical manifestations and the inadequacy of conventional bacterial identification methods [5,6,8,9].

Studies from Australia and South East Asia indicate that environmental and host factors determine disease acquisition. The disease incidence increases during the rainy season and adverse weather conditions like tsunamis and cyclones; agricultural workers are commonly affected [6,8,11,12]. Host factors such as the presence of one or more preexisting conditions that alter immune response (such as long standing uncontrolled diabetes mellitus or chronic renal failure), severe or penetrating injury or near-drowning are favorable for the occurrence of melioidosis [8,12,13]. In India, cases of melioidosis have been recognized from different regions, however case identification is confined to few tertiary centres due to limited diagnostic facilities [14–16]. The incidence of melioidosis in India is unknown but could be substantial due to the high burden of diabetes mellitus [17] and long coastline prone to extreme weather conditions.

On July 23, 2018, the death of a boy aged 17 years was reported to Moodabettu Primary Health Centre (PHC), Udupi district of Karnataka. On July 24, a team from the district disease surveillance office visited hospital A, where the deceased received treatment and had been diagnosed with melioidosis. The surveillance team also searched for similar cases in the village and conducted an awareness program for reporting sudden deaths. National Centre for Disease Control (NCDC) was notified, and Epidemic Intelligence Service Officers (EISOs) from NCDC joined the district investigation on August 1, 2018. We investigated to describe the epidemiology and identify risk factors to inform the initial public health responses.

## Methods

### I. Ethics statement

This investigation was undertaken as part of an emergency public health response to identify the cause of an outbreak for early intervention. All statutory permissions were obtained from NCDC and Integrated Disease Surveillance Programme. Ethical approval was exempted as the investigation was conducted consistent with applicable state and central government law (Epidemic Diseases Act no.3, 1897). Strict data protection protocols reviewed by NCDC were followed while collecting information from cases and controls.

### II. Case investigation

To ascertain the cause of death, we interviewed the treating physicians and family members of the deceased. Information was collected on clinical presentation, the timeline of events before the death, treatment and laboratory results. The patient's CSF specimen was plated on routine media as well as in BacT/ALERT automated culture system (bioMérieux, Marcy-L'Etoile, France). The patient's blood was also collected to rule out septicemia and cultured in BacT/ALERT automated culture system. The preliminary Gram stain of the CSF sample revealed bipolar gram negative staining leading to a presumptive diagnosis, which was conveyed to the treating doctors to initiate immediate empirical treatment. The culture isolate from CSF after 48 hours of incubation at 37˚C with 5% $CO_2$ were examined by latex agglutination, matrix assisted laser desorption ionization-time of flight mass spectrometry (MALDI-TOF), and type three secretion system (TTSS) polymerase chain reaction (PCR).

### III. Descriptive epidemiology

**Study site.** Udupi district has a population of 1,177,908 and a literacy rate of 83% [18]. The weather is hot and humid during summers and receives heavy rainfall from May-August.

The population's comprises of agricultural communities, and rice is the main crop grown in the region. Udupi district has 76 primary health centres, six community health centres and a district hospital in the urban area. Patients who are critically ill are referred from most of Udupi district, other parts of Karnataka state and nearby districts of Kerala state to hospital A, which is a tertiary care teaching hospital.

**Case definition for descriptive analysis.** A case was defined as a diagnosis of culture-confirmed melioidosis (isolation of *B*. *pseudomallei* from any clinical sample and suggestive clinical features) in a patient from the Udupi district who was admitted to the ward/intensive care unit of hospital A from January 2013—July 2018.

**Case search and data collection.** We reviewed inpatient records and laboratory registers from 2013–18 from hospital A. For patients meeting the case definition, we collected clinical and laboratory data and created a line list without patient identifiers. Clinical data was available for patients attending the hospital after 2016.

## IV. Case-control study

We conducted a 1:3 matched case-control study to identify risk factors on hospitalized patients meeting the above case definition from January 2017 to July 2018 to limit recall bias. A control was defined as a resident of Udupi district admitted to hospital A in July (rainy season) 2018 with a non-infectious condition. We matched each case with three hospital controls by age and sex group (males age 15–43 years, males >50 years and females >49 years). All controls were selected from the hospital admission records (eligible controls were listed and selected by random number generation using MS Excel using RANDBETWEEN function). We excluded patients receiving antibiotic treatment for pneumonia or sepsis.

A semi-structured questionnaire was used to collect data on socio-demographics, housing conditions, water storage practices, daily living activities, injuries, and environmental exposures (three months before hospitalization) from cases and controls during interviews. For the deceased cases, we conducted proxy interviews with family members.

**Data analysis.** The Epi Info software 7.2 was used to analyze frequencies and proportions. For the identified risk factors, crude odds ratio (OR) with a 95% confidence interval (CI) was calculated. The exact conditional logistic regression analysis was run to obtain matched odds ratios (mOR) and 95% CI considering the small sample size using SAS version 9.4.

## Results

### I. Case investigation

A 17-year-old boy from the Udupi district developed complaints of fever and vomiting on July 7, 2018. On July 8, his illness progressed by evening, and he developed high-grade fever, severe headache and dizziness. On July 9, the family noticed deviation at the angle of the mouth and consulted a local doctor. The boy's condition deteriorated and he was taken to hospital A, where he was admitted on July 11 with difficulty in swallowing, change in voice and ataxia. He developed seizures and coma on July 14 and was transferred to intensive care. His condition deteriorated with neurological involvement, and he died on July 21 in hospital A. The patient had a history of fall on June 25 during a Kabaddi game (Kabaddi is a local team game of chasing the opponent team played barefoot on a muddy playground). No external injuries were apparent, but minor abrasions on limbs and hands were noted. The case had no history of diabetes, renal dysfunction, or other comorbidities. He was diagnosed with neuro-melioidosis by cerebrospinal fluid culture and PCR. His blood culture for bacteria including mycobacterium tuberculosis was negative. Contrast magnetic resonance imaging of the brain and spinal cord showed patchy T2 and FLAIR hyperintense lesions in the brainstem, middle cerebellar

peduncles, and bilateral posterior limbs of internal capsule, in addition to few subcortical lesions. Post-contrast enhancement was seen in the brainstem lesions, along the trigeminal nerves, facial and abducens nuclei. This case was later reported as part of case series showing spectrum of nervous involvement in melioidosis with detailed clinical description [19].

## II. Descriptive epidemiology of cases

A total of 50 cases of melioidosis were identified at hospital A from 2013–18. The median age of cases was 52 (range 17–83) years; 80% were males, and the case fatality rate was 16%. The cases were distributed (by residence) in all the three taluks (administrative subdivisions) of the Udupi district (Fig 1); 76% of cases occurred in the villages closer to the coastline, and 66% of cases occurred between May and October (Fig 2).

We analyzed the clinical presentation of all 19 cases reported from January 2017- July 2018. The most common presenting symptoms were fever (89%), cough (42%), joint pain (37%) and abscess (16%). Uncontrolled diabetes (HbA1c >7%) was documented in 84% (16) cases with overall median HbA1c of 9.5% (range 5.8% -13%). The other chronic comorbidities included chronic kidney disease with diabetes (10%), COPD with diabetes (10%), liver disease with diabetes (10%), tuberculosis (5%), and cancer (5%). Among 19 cases, the most common presentation was bacteremic melioidosis (58%), followed by skin and soft tissue (11%), septic arthritis (11%), pneumonia (5%), splenic abscess (5%), neuromelioidosis (5%) and no focus (5%). Out of 19 cases, five had lung involvement; of these, one had focal lung involvement and four others had lung involvement with bacteremia. All five cases with pneumonia were exposed either to soil or stagnant water. Among the 14 non pneumonia cases, 13 had history of exposure to soil or stagnant water.

## III. Case-control study

We enrolled all 19 cases from 2017–18 and 57 hospital controls in the matched cases control analysis. The enrollment flowchart for study participants is in appendix page 1. Univariate analysis showed that melioidosis cases were more likely than controls to have an injury with breach of skin, contact with stagnant water, wet soil, and both. In matched analysis, injury with a breach in skin and contact with stagnant water had a significant association with illness. In addition, we also looked for activity-specific odds ratios for contact with stagnant water. The odds of exposure to swimming in stagnant water, working in paddy field, and walk in waterlogged areas were higher among cases compared to controls (Table 1).

## Discussion

The death of an adolescent due to neuromelioidosis led to a broader, multi-year retrospective investigation of admitted cases of melioidosis in one district. These cases were mostly males with uncontrolled diabetes. The results from the case-control study suggest that outdoor exposure to stagnant water and wet soil in rainy season are a risk factor for melioidosis in the Udupi district.

We observed that the majority (66%) of the cases occurred during the rainy session (May-October). This is similar to the seasonality reported from the west coastal region of India, Singapore and Australia [11,16,21]. The descriptive analysis showed most (76%) cases were from villages closer to the coast with paddy fields and low-lying areas that frequently flood during the rainy session. These environmental conditions could increase the chances of contact with contaminated wet soil and stagnant water during agricultural and non-agricultural activities associated with infection. In our case-control study, exposure to wet soil and stagnant water were significant risk factors for melioidosis (p<0.001). Additionally, we also found that

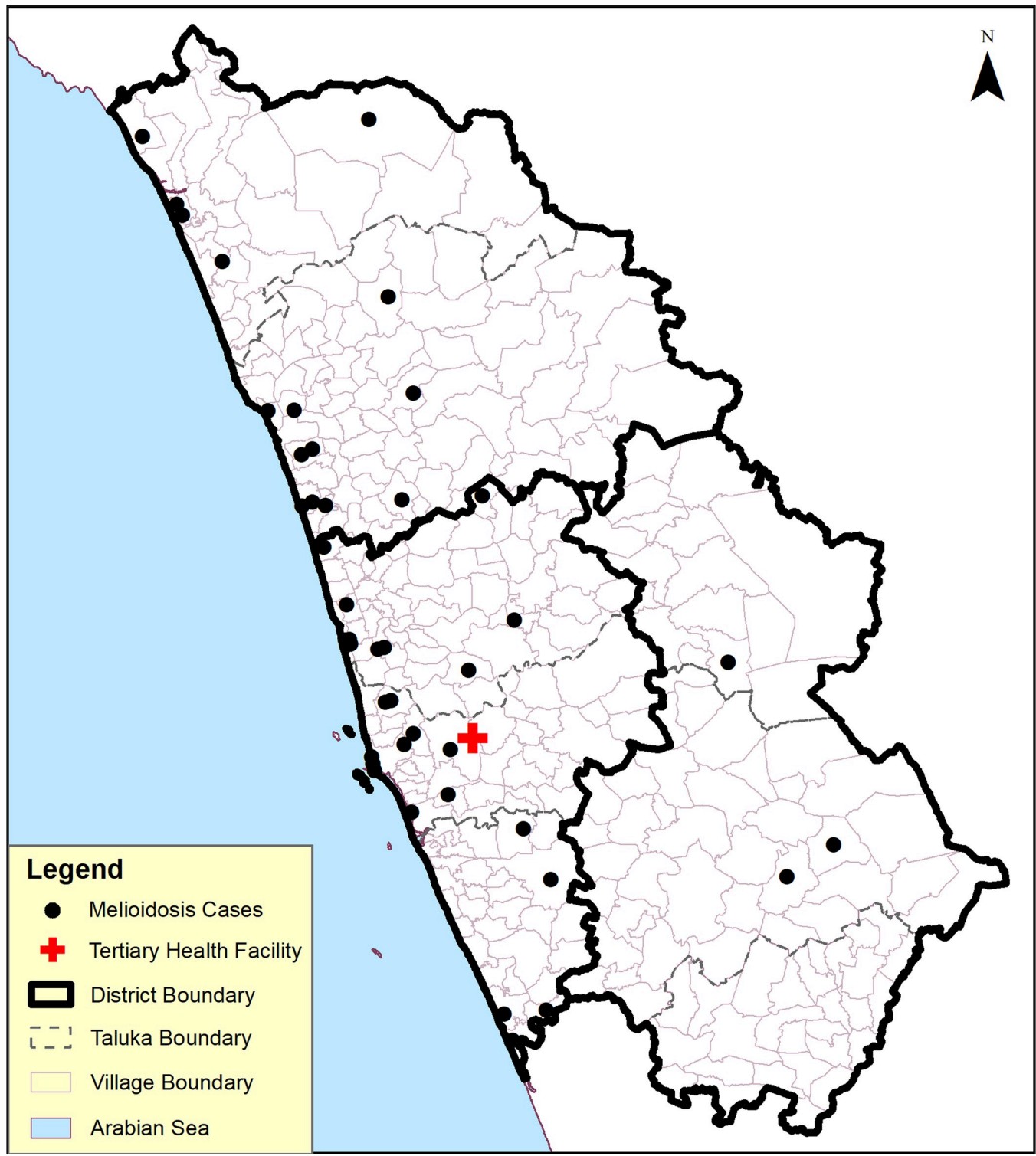

**Fig 1. Geographical distribution of melioidosis cases from 2013–2018 Udupi District, Karnataka, India (n = 50).** Base map republished from [20] under a CC BY license, with permission from Karnataka State Remote Sending Application Centre (KSRC), original copyright KSRC, 2022.

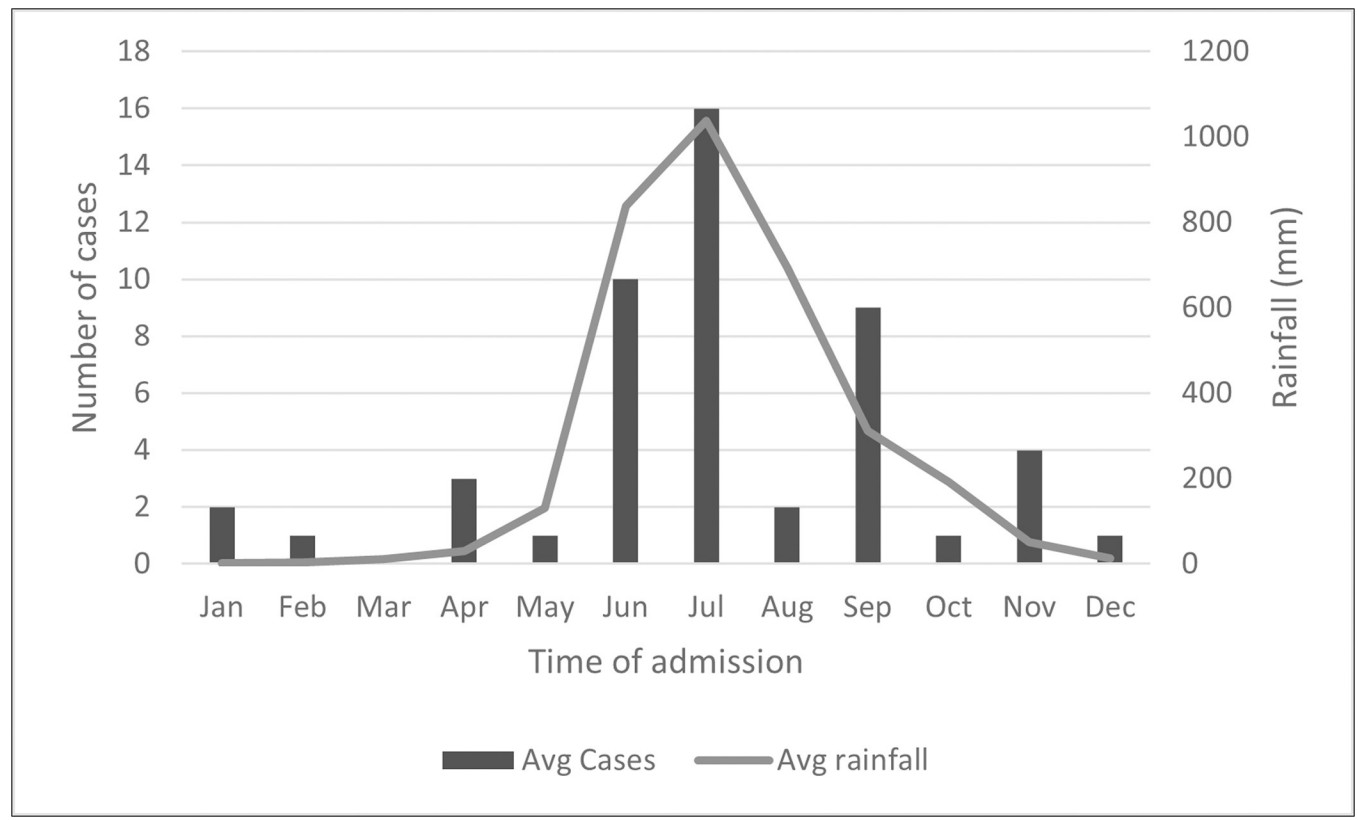

**Fig 2. Monthly distribution of average melioidosis cases and rainfall from 2013–2018, Udupi district, Karnataka, India (n = 50).**

sustaining cut injuries was an independent predictor in multivariate analysis. Melioidosis cases were six times more likely to be exposed to cut injuries compared to controls. These findings, combined with environmental conditions, indicate that percutaneous inoculation is an important transmission mode for melioidosis in rural India. Similar epidemiological risk factors were identified among melioidosis cases in rural Thailand and northern Australia [11,12]. In contrast, recent findings have also recognized inhalation of *B. pseudomallei* and eating food contaminated with soil or dust as other important transmission modes [21,22].

People with long-standing diabetes with poor glycemic control are known to be at increased risk of acquiring melioidosis [6,12], and 84% of the cases in this study detected between 2013–2018 had uncontrolled diabetes. A systematic review from India for the period 1991–2018 found diabetes mellitus to be a major predisposing condition in 70% of reported cases [16]. There are multiple reports of melioidosis from various parts of India [15,16,23–25], but it is not a notifiable disease. Estimates suggest that melioidosis is endemic to India with an annual burden of ~52,006 cases and death count of 31,425 (13,405–75,601) cases, with further escalation of the mortality rate to 90% if the disease remains undiagnosed and untreated [10]. Given the high burden of diabetes in India, inadequate diagnostic facilities in the microbiology laboratories, especially in rural parts [26] and low awareness among physicians and the public, the actual burden of melioidosis in India is expected to be high. Therefore, priorities include establishing state and national referral laboratory networks, training for diagnosis and surveillance for melioidosis, and increasing awareness in the community and among physicians.

Though neuromelioidosis is a rare (4–5%) presentation of melioidosis [19,27,28], children and adolescents have poor outcomes, frequently resulting in either death or neurological

**Table 1. Risk factors associated with melioidosis in Udupi district 2017–2018.**

| Risk Factors | Case (n = 19) | | Control (n = 57) | | OR | (95% CI) | mOR (95% CI) | | p-value |
|---|---|---|---|---|---|---|---|---|---|
| | n | (%) | n | (%) | | | | | |
| **Injury leading to breach of skin** | 10 | (53) | 9 | (16) | 5.9 | (1.9–18.7) | 6.2 | (1.9–20.1) | 0.0014 |
| **Multiple chronic comorbidities††** | 4 | (21) | 6 | (10) | 2.3 | (0.6–9.1) | 2.2 | (0.6–8.9) | 0.2464 |
| **Contact with stagnant water‡** | 7 | (37) | 4 | (7) | 63.0 | (6.1–651) | 58.2 | (5.2->999) | <0.001 |
| **Contact with wet soil†** | 3 | (16) | 8 | (14) | 13.5 | (1.2–147) | 10.8 | (0.7–630) | 0.0887 |
| **Contact with stagnant water and wet soil** | 8 | (42) | 9 | (16) | 32.0 | (3.5–290) | 28.2 | (3.2->999) | 0.0004 |
| **No contact with wet soil or stagnant water** | 1 | (5) | 36 | (63) | Ref | | Ref | | |
| **Contact with stagnant water** | | | | | | | | | |
| Swimming in pond, work in paddy field and walk in waterlogged area | 6 | (32) | 0 | (0) | 93.0 | (8.4–4030) | >999 | (<0->999) | 0.9941 |
| Swimming in pond and work in paddy field | 3 | (16) | 2 | (3) | 29.0 | (2.9–287) | 48.6 | (4.3–556) | 0.0018 |
| Swimming in pond and walk in waterlogged area | 2 | (10) | 2 | (3) | 19.5 | (1.7–219) | 16.9 | (1–198) | 0.0247 |
| Work in paddy field | 2 | (10) | 5 | (9) | 7.8 | (0.9–68) | 12.2 | (1.2–126) | 0.0362 |
| Swimming in pond | 2 | (10) | 5 | (9) | 7.8 | (0.9–68) | 11.2 | (1.1–110) | 0.0379 |
| Walk in waterlogged area | 2 | (10) | 4 | (8) | 9.7 | (1.1–89) | 10.1 | (1.1–96) | 0.0438 |
| No contact with stagnant water | 2 | (10) | 39 | (68) | Ref | | Ref | | |

OR: Odds Ratio, mOR: Matched Odds Ratio by Exact Conditional Logistic Regression Analysis and p-value by conditional likelihood estimates.

†† More than one chronic comorbidity reported (Diabetes mellitus, liver disease, chronic kidney disease, tuberculosis, COPD, cancers, HIV, cystic fibrosis, any other immunocompromised state).

† Composite variable created after combining responses for swimming in the pond (yes), work in paddy field (yes), and walk in waterlogged areas (yes).

‡ Composite variable created after combining responses for work as construction labour (yes), farming (yes), and gardening (yes).

impairment (37%) attributed to severe sepsis and its complications, resulting from delay in treatment [5,28]. Studies from Australia [29] and Cambodia [30] have documented a case fatality rate of 7–17%; notably children in the Australian study did not have any underlying comorbidities, similar to the young index patient in our study.

Our study has limitations, including recall bias for exposure due to the retrospective nature of the study design. We attempted to limit this by enrolling the most recent cases and asking for exposures from cases and controls within a three-month reference period. The enrollment period for cases and controls was different due to logistical issues. However, we ensured comparability of risks by enrolling controls during the rainy season, as more than half of the cases were reported during this season. Finally, the findings are from admitted cases from a single hospital were enrolled, potentially limiting the generalizability of our findings.

We recommended enhancing the education of medical doctors for early diagnosis and treatment of melioidosis among fever cases; and strengthening district public health labs in Karnataka state for melioidosis diagnosis. We also recommended communication campaigns targeting high-risk groups (diabetics in coastal areas, patients with chronic renal failures, agricultural workers and people with frequent soil exposure like daily labourers). These communication campaigns should focus on minimizing unnecessary exposures to soil and water through avoidance or protective clothing including use of footwear.

Following this investigation, the National Centre for Disease Control published an information bulletin to increase awareness among clinicians of melioidosis as a differential diagnosis in fever of unknown origin and community acquired pneumonia. To enhance surveillance, an expert group meeting was convened to develop surveillance case definitions, and district lab personnel training was initiated to strengthen melioidosis diagnostic capacity.

## Supporting information

**S1 Fig. Flow diagram showing selection of melioidosis cases, in the case control study, Udupi District, Karnataka.**
(DOCX)

## Author Contributions

**Conceptualization:** Akhileshwar Singh, Ashok Talyan, Anubhav Srivastav, Chiranjay Mukhopadhyay, Suma Nair, Mohan Papanna, Rajesh Yadav, Sujeet Kumar Singh, Tanzin Dikid.

**Data curation:** Ashok Talyan, Vasudeva Upadhya, Chiranjay Mukhopadhyay, Suma Nair, Mohan Papanna, Rajesh Yadav, Sujeet Kumar Singh.

**Formal analysis:** Akhileshwar Singh, Ashok Talyan, Chiranjay Mukhopadhyay, Mohan Papanna, Rajesh Yadav, Tanzin Dikid.

**Investigation:** Akhileshwar Singh, Ramesh Chandra, Shyamsundar Shreedhar, Deepak Sudhakaran, Suma Nair.

**Methodology:** Akhileshwar Singh, Ramesh Chandra, Tanzin Dikid.

**Supervision:** Tanzin Dikid.

**Writing – original draft:** Mohan Papanna, Rajesh Yadav, Tanzin Dikid.

**Writing – review & editing:** Chiranjay Mukhopadhyay, Mohan Papanna, Rajesh Yadav, Tanzin Dikid.

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
