## [Decision Letter · Decision Letter 0]

7 Jun 2022

PGPH-D-22-00623

Risk Factors for Melioidosis in Udupi District, Karnataka, India, August 2018

Dear Dr. Dikid,

Thank you for submitting your manuscript to PLOS Global Public Health. After careful consideration, we feel that it has merit but does not fully meet PLOS Global Public Health’s publication criteria as it currently stands. Therefore, we invite you to submit a revised version of the manuscript that addresses the points raised during the review process.

Please submit your revised manuscript by . If you will need more time than this to complete your revisions, please reply to this message or contact the journal office at globalpubhealth@plos.org. Please include the following items when submitting your revised manuscript:

We look forward to receiving your revised manuscript.

Kind regards,

Raquel Muñiz-Salazar, Ph.D.

Academic Editor

Journal Requirements:

1. Please provide  separate figure files in .tif or .eps format only and remove any figures embedded in your manuscript file.  Please ensure that all files are under our size limit of 20MB.  

For more information about how to convert your figure files please see our guidelines: Once you've converted your files to .tif or .eps, please also make sure that your figures meet our format requirements:

2. Please update the 'Competing Interests' statement in the system with "The authors have declared that no competing interests exist".

3. Please amend your detailed Financial Disclosure statement. This is published with the article. It must therefore be completed in full sentences and contain the exact wording you wish to be published.

4. Figures 1 and 2: please (a) provide a direct link to the base layer of the map used and ensure this is also included in the figure legend; (b) provide a link to the terms of use / license information for the base layer. We cannot publish proprietary or copyrighted maps (e.g. Google Maps, Mapquest) and the terms of use for your map base layer must be compatible with our CC-BY 4.0 license. 

5. Please amend your Data Availability Statement and indicate where the data may be found

Additional Editor Comments (if provided):

Three reviewers agree that this is a critical manuscript highlighting the need for public health intervention to decrease melioidosis incidence in India. It is essential to improve the manuscript by attending to the reviewers' comments.

REVIEWER 1

Overall this is an important manuscript highlighting the need for public health intervention to decrease the incidence of melioidosis in India. I believe this manuscript could be improved and have included my comments below.

Major:

Line 149: please explain why only 19 of the 50 cases were analyzed. It might be best to include a flow diagram with inclusions and exclusions for clarity.

Line 202: there is no mention of severity in the case definition, but is mentioned as a factor in the limitation sections. Please clarify.

It has been mentioned that melioidosis is not a notifiable condition in India. There are a number of instances in the manuscript where the term “reported” or “reporting” has been used, for example in Figure 2. For clarity are these the cases identified by the investigation or the cases sporadically reported to the NCDC?

There is no mention of additional investigations relating to the case. Did he have pneumonia? Was he bacteremic? Was a swab of the abrasion site taken, and if so was it culture positive for B. pseudomallei?

There is a very limited description of patient presentation: “The most common type was bacteremic melioidosis (53%).” This should be expanded, preferably in a table, to include: bacteremia, pneumonia, skin & soft tissue, organ involvement (i.e. liver abscess etc.), neuromelioidosis, no focus.

What was the difference between those with pneumonia vs no pneumonia in the context of the various environmental risk factors? It would be important to mention this information as inhalation is a common route of acquisition.

What was the predominant presentation in the dry season compared to the wet season?

Table 1 should include the specific risk factors included in the results section.

Minor:

Title: “August 2018” suggests that only this month was assessed. Consider rephrasing.

Line 86: I assume the Gram stain revealed bipolar Gram negative staining?

Line 88: incubated in room air/aerobic conditions or in 5% CO2?

Line 93: reference

Data analysis section: consider removing “we”

Line 133: transferred to intensive care

Line 139 – 141: suggest rewrite

Line 154: neuromeliodosis should not be counted as a chronic comorbidity

Line 154: “the most common type” suggest rewriting this sentence.

Line 165: “and having” with

Line 205: this sentence is misleading as there is no definition of fever of unknown origin mentioned in this manuscript. In this context it is more likely that the disease is either misdiagnosed or undiagnosed, and not a fever of unknown origin (e.g. a fever that lasts for >3 weeks or failure to reach a diagnosis after 1 week of appropriate investigations)

Line 206 – 209: grammar

Figure 2: it would be great if rainfall data could be included in this figure to demonstrate the correlation with the wet season.

REVIEWER 2

Authors have investigated the risk factors for acquiring melioidosis in Udupi district in India. It will be helpful in future implementations of control and preventive strategies for melioidosis.

My specific comments are as follows.

Major point:

Point 1: Line 202: “However, we tried to ensure comparability of exposures by enrolling controls during the rainy season, a time of high risk” It is not clear why authors tried to ensure comparability of exposure. Why were the controls not distributed throughout the year, as it would have brought out the seasonal risk factor, and risk factors associated with exposure better. Authors should not have tried to match the exposure.

Point 2: Line 139-141: The MRI description is not accurate. It cannot be “right side” of the corticospinal tract. Authors are directed to quote the case report which describes the same case from the region elaborately with MRI pictures in a previous publication that was published by the clinicians involved in case management- Chatterjee A, Saravu K, Mukhopadhyay C, Chandran V. Neurological Melioidosis Presenting as Rhombencephalitis, Optic Neuritis, and Scalp Abscess with Meningitis: A Case Series from Southern India. Neurol India. 2021 Mar-Apr;69(2):480-482. doi: 10.4103/0028-3886.314590. PMID: 33904481

Minor points:

Line 65-66: In India, sporadic case reports of melioidosis have been mostly from Karnataka and Tamil Nadu due to better diagnostic capacity in these regions. Authors fail to quote this paper which has for the first time described a case series of 25 patients from the same region as study setting, which has also described seasonal preponderance. Saravu K, Mukhopadhyay C, Vishwanath S, Valsalan R, Docherla M, Vandana KE, Shastry BA, Bairy I, Rao SP. Melioidosis in southern India: epidemiological and clinical profile. Southeast Asian J Trop Med Public Health. 2010 Mar;41(2):401-9. PMID: 20578524.

Line 154: Why is neuromeliodosis written as “other chronic co morbidity”? It is the very subject of investigation.

Grammar and Typos are to be corrected in the manuscript

Line 184. A systematic review from India for the period 1991-2018 found diabetes mellitus (to be ) a major predisposing condition in 70% of reported cases

Line 193-194 “children and adolescents have poor outcomes, frequently resulting in either death or neurological impairment (37%) (5, 25)”. What is the possible explanation for this finding?

Line 315: Muthusamy KA, Waran V, Puthucheary SD. Spectra of central nervous system melioidosis. J Clin 316 Neurosci. 2007;14:1213-5. A very old sole reference (2007) has been quoted while there are newer reports of neuromeliodosis.

Newer references and unique presentations that are reported recently(2015 and 2021) may be included for neuromeliodosis.

1.Chatterjee A, Saravu K, Mukhopadhyay C, Chandran V. Neurological Melioidosis Presenting as Rhombencephalitis, Optic Neuritis, and Scalp Abscess with Meningitis: A Case Series from Southern India. Neurol India. 2021 Mar-Apr;69(2):480-482. doi: 10.4103/0028-3886.314590. PMID: 33904481

2. Saravu K, Kadavigere R, Shastry AB, Pai R, Mukhopadhyay C. Neurologic melioidosis presented as encephalomyelitis and subdural collection in two male labourers in India. J Infect Dev Ctries. 2015 Nov 30;9(11):1289-93. doi: 10.3855/jidc.6586. PMID: 26623640.

1) Page 5 - Abstract. - The methodology seems to be misleading as only 19 cases underwent logistic matched case-control analysis. Yet, you have stated that 50 cases were identified. If I am not wrong, this maybe the incident cases for the retrospective epidemiological survey, all of which were not included in the statistical analysis. Please consider clarifying how many cases underwent the specified analysis.

2) Page 6 , Introduction - Line 64-66. I have to disagree with the statement that melioidosis reporting was limited to Karnataka and Tamil Nadu due to diagnostic limitations in other south-indian states in general. I believe diagnostic limitation may instead be tertiary care centres, that most south indian states do feature albeit far and between and due to the general unawareness of the condition. Citations no. 14 and 15 are quite outdated. I recommend citing more recent published reports from other states such as https://doi.org/10.1155/2021/8154810, https://doi.org/10.3390/idr12030011 or even our published paper from https://doi.org/10.1155/2021/8154810.

3) Page 9 - Results - I - Case investigation - It would have been "nice" to have a MRI image of the patient with neuro-melioidosis - Completely optional but can improve reader engagement.

4) Page 10 - Descriptive epidemiology - Only minimal baselines information is given on the 50 cases identified from 2013 to 2018. It would have been helpful to delve deep into the clinical presentation and cause of mortality of these patients ( and not just the 19), as data is probably already available from hospital record. Since objective evidence is available, I doubt recall bias would have limited the descriptive analysis.

5) Page 11 - III- Case control study - Since there was no minimum sample size calculation, I am doubtful of the statistical power the regression analysis contains due to the relatively small sample analysed (wide confidence inferences noted for risk factors, many with statistical non-significance at alpha of 0.05). I recommend adding the power analysis and goodness-of-fit report, as it may inform the readers on the strength of the inferences made as the effect sizes (mOR) are quite large.

6) The discussion is well drafted with a succinct comparison with existing literature, pragmatic suggestions to policy makers, with strengths and weaknesses of the study described in detail.

Reviewer no. 3

Reviewers' comments:

Reviewer's Responses to Questions

**Comments to the Author**

1. Does this manuscript meet PLOS Global Public Health’s publication criteria? Is the manuscript technically sound, and do the data support the conclusions? The manuscript must describe methodologically and ethically rigorous research with conclusions that are appropriately drawn based on the data presented.

Reviewer #1: Partly

Reviewer #2: Yes

Reviewer #3: Partly

2. Has the statistical analysis been performed appropriately and rigorously?

Reviewer #1: Yes

Reviewer #2: Yes

Reviewer #3: No

3. Have the authors made all data underlying the findings in their manuscript fully available (please refer to the Data Availability Statement at the start of the manuscript PDF file)?

Reviewer #1: Yes

Reviewer #2: Yes

Reviewer #3: No

4. Is the manuscript presented in an intelligible fashion and written in standard English?

Reviewer #1: Yes

Reviewer #2: Yes

Reviewer #3: Yes

5. Review Comments to the Author

Reviewer #1: Overall this is an important manuscript highlighting the need for public health intervention to decrease the incidence of melioidosis in India. I believe this manuscript could be improved and have included my comments below.

Major:

Line 149: please explain why only 19 of the 50 cases were analyzed. It might be best to include a flow diagram with inclusions and exclusions for clarity.

Line 202: there is no mention of severity in the case definition, but is mentioned as a factor in the limitation sections. Please clarify.

It has been mentioned that melioidosis is not a notifiable condition in India. There are a number of instances in the manuscript where the term “reported” or “reporting” has been used, for example in Figure 2. For clarity are these the cases identified by the investigation or the cases sporadically reported to the NCDC?

There is no mention of additional investigations relating to the case. Did he have pneumonia? Was he bacteremic? Was a swab of the abrasion site taken, and if so was it culture positive for B. pseudomallei?

There is a very limited description of patient presentation: “The most common type was bacteremic melioidosis (53%).” This should be expanded, preferably in a table, to include: bacteremia, pneumonia, skin & soft tissue, organ involvement (i.e. liver abscess etc.), neuromelioidosis, no focus.

What was the difference between those with pneumonia vs no pneumonia in the context of the various environmental risk factors? It would be important to mention this information as inhalation is a common route of acquisition.

What was the predominant presentation in the dry season compared to the wet season?

Table 1 should include the specific risk factors included in the results section.

Minor:

Title: “August 2018” suggests that only this month was assessed. Consider rephrasing.

Line 86: I assume the Gram stain revealed bipolar Gram negative staining?

Line 88: incubated in room air/aerobic conditions or in 5% CO2?

Line 93: reference

Data analysis section: consider removing “we”

Line 133: transferred to intensive care

Line 139 – 141: suggest rewrite

Line 154: neuromeliodosis should not be counted as a chronic comorbidity

Line 154: “the most common type” suggest rewriting this sentence.

Line 165: “and having” with

Line 205: this sentence is misleading as there is no definition of fever of unknown origin mentioned in this manuscript. In this context it is more likely that the disease is either misdiagnosed or undiagnosed, and not a fever of unknown origin (e.g. a fever that lasts for >3 weeks or failure to reach a diagnosis after 1 week of appropriate investigations)

Line 206 – 209: grammar

Figure 2: it would be great if rainfall data could be included in this figure to demonstrate the correlation with the wet season.

Reviewer #2: Authors have investigated the risk factors for acquiring melioidosis in Udupi district in India. It will be helpful in future implementations of control and preventive strategies for melioidosis.

My specific comments are as follows.

Major point:

Point 1: Line 202: “However, we tried to ensure comparability of exposures by enrolling controls during the rainy season, a time of high risk” It is not clear why authors tried to ensure comparability of exposure. Why were the controls not distributed throughout the year, as it would have brought out the seasonal risk factor, and risk factors associated with exposure better. Authors should not have tried to match the exposure.

Point 2: Line 139-141: The MRI description is not accurate. It cannot be “right side” of the corticospinal tract. Authors are directed to quote the case report which describes the same case from the region elaborately with MRI pictures in a previous publication that was published by the clinicians involved in case management- Chatterjee A, Saravu K, Mukhopadhyay C, Chandran V. Neurological Melioidosis Presenting as Rhombencephalitis, Optic Neuritis, and Scalp Abscess with Meningitis: A Case Series from Southern India. Neurol India. 2021 Mar-Apr;69(2):480-482. doi: 10.4103/0028-3886.314590. PMID: 33904481

Minor points:

Line 65-66: In India, sporadic case reports of melioidosis have been mostly from Karnataka and Tamil Nadu due to better diagnostic capacity in these regions. Authors fail to quote this paper which has for the first time described a case series of 25 patients from the same region as study setting, which has also described seasonal preponderance. Saravu K, Mukhopadhyay C, Vishwanath S, Valsalan R, Docherla M, Vandana KE, Shastry BA, Bairy I, Rao SP. Melioidosis in southern India: epidemiological and clinical profile. Southeast Asian J Trop Med Public Health. 2010 Mar;41(2):401-9. PMID: 20578524.

Line 154: Why is neuromeliodosis written as “other chronic co morbidity”? It is the very subject of investigation.

Grammar and Typos are to be corrected in the manuscript

Line 184. A systematic review from India for the period 1991-2018 found diabetes mellitus (to be ) a major predisposing condition in 70% of reported cases

Line 193-194 “children and adolescents have poor outcomes, frequently resulting in either death or neurological impairment (37%) (5, 25)”. What is the possible explanation for this finding?

Line 315: Muthusamy KA, Waran V, Puthucheary SD. Spectra of central nervous system melioidosis. J Clin 316 Neurosci. 2007;14:1213-5. A very old sole reference (2007) has been quoted while there are newer reports of neuromeliodosis.

Newer references and unique presentations that are reported recently(2015 and 2021) may be included for neuromeliodosis.

1.Chatterjee A, Saravu K, Mukhopadhyay C, Chandran V. Neurological Melioidosis Presenting as Rhombencephalitis, Optic Neuritis, and Scalp Abscess with Meningitis: A Case Series from Southern India. Neurol India. 2021 Mar-Apr;69(2):480-482. doi: 10.4103/0028-3886.314590. PMID: 33904481

2. Saravu K, Kadavigere R, Shastry AB, Pai R, Mukhopadhyay C. Neurologic melioidosis presented as encephalomyelitis and subdural collection in two male labourers in India. J Infect Dev Ctries. 2015 Nov 30;9(11):1289-93. doi: 10.3855/jidc.6586. PMID: 26623640.

Reviewer #3: Thank you for the opportunity to review this scientific article. The article seems to present a case report of a recent mortality with neuro-melioidosis and further presents a case-control analysis of 19 cases between 2013 and 2018. This is an important epidemiological survey on a neglected tropical disease, which is all the more likely to exacerbate due to climate change and natural calamities. I have a few concerns that when addressed could potentially improve the manuscript.

1) Page 5 - Abstract. - The methodology seems to be misleading as only 19 cases underwent logistic matched case-control analysis. Yet, you have stated that 50 cases were identified. If I am not wrong, this maybe the incident cases for the retrospective epidemiological survey, all of which were not included in the statistical analysis. Please consider clarifying how many cases underwent the specified analysis.

2) Page 6 , Introduction - Line 64-66. I have to disagree with the statement that melioidosis reporting was limited to Karnataka and Tamil Nadu due to diagnostic limitations in other south-indian states in general. I believe diagnostic limitation may instead be tertiary care centres, that most south indian states do feature albeit far and between and due to the general unawareness of the condition. Citations no. 14 and 15 are quite outdated. I recommend citing more recent published reports from other states such as https://doi.org/10.1155/2021/8154810, https://doi.org/10.3390/idr12030011 or even our published paper from https://doi.org/10.1155/2021/8154810.

3) Page 9 - Results - I - Case investigation - It would have been "nice" to have a MRI image of the patient with neuro-melioidosis - Completely optional but can improve reader engagement.

4) Page 10 - Descriptive epidemiology - Only minimal baselines information is given on the 50 cases identified from 2013 to 2018. It would have been helpful to delve deep into the clinical presentation and cause of mortality of these patients ( and not just the 19), as data is probably already available from hospital record. Since objective evidence is available, I doubt recall bias would have limited the descriptive analysis.

5) Page 11 - III- Case control study - Since there was no minimum sample size calculation, I am doubtful of the statistical power the regression analysis contains due to the relatively small sample analysed (wide confidence inferences noted for risk factors, many with statistical non-significance at alpha of 0.05). I recommend adding the power analysis and goodness-of-fit report, as it may inform the readers on the strength of the inferences made as the effect sizes (mOR) are quite large.

6) The discussion is well drafted with a succinct comparison with existing literature, pragmatic suggestions to policy makers, with strengths and weaknesses of the study described in detail.

6. PLOS authors have the option to publish the peer review history of their article (what does this mean?). If published, this will include your full peer review and any attached files.

**Do you want your identity to be public for this peer review?** For information about this choice, including consent withdrawal, please see our Privacy Policy.

Reviewer #1: No

Reviewer #2: No

Reviewer #3: **Yes: **Manu Pradeep

---

## [Decision Letter · Decision Letter 1]

26 Sep 2022

PGPH-D-22-00623R1

Risk Factors for Melioidosis in Udupi District, Karnataka, India, January 2017-July 2018

Dear Dr. Dikid,

Thank you for submitting your manuscript to PLOS Global Public Health. After careful consideration, we feel that it has merit but does not fully meet PLOS Global Public Health’s publication criteria as it currently stands. Therefore, we invite you to submit a revised version of the manuscript that addresses the points raised during the review process.

We look forward to receiving your revised manuscript.

Kind regards,

Raquel Muñiz-Salazar, Ph.D.

Academic Editor

Journal Requirements:

2. Please provide a/amend your detailed Financial Disclosure statement. This is published with the article. It must therefore be completed in full sentences and contain the exact wording you wish to be published.

If you did not receive any funding for this study, please simply state: “The authors received no specific funding for this work."

Additional Editor Comments (if provided):

Dear author.

Because all comments have been addressed satisfactorily, the reviewers have decided to accept the manuscript after a Minor Revision.

Reviewers' comments:

Reviewer's Responses to Questions

**Comments to the Author**

1. If the authors have adequately addressed your comments raised in a previous round of review and you feel that this manuscript is now acceptable for publication, you may indicate that here to bypass the “Comments to the Author” section, enter your conflict of interest statement in the “Confidential to Editor” section, and submit your "Accept" recommendation.

Reviewer #2: All comments have been addressed

Reviewer #4: (No Response)

2. Does this manuscript meet PLOS Global Public Health’s publication criteria? Is the manuscript technically sound, and do the data support the conclusions? The manuscript must describe methodologically and ethically rigorous research with conclusions that are appropriately drawn based on the data presented.

Reviewer #2: Yes

Reviewer #4: Yes

3. Has the statistical analysis been performed appropriately and rigorously?

Reviewer #2: Yes

Reviewer #4: (No Response)

4. Have the authors made all data underlying the findings in their manuscript fully available (please refer to the Data Availability Statement at the start of the manuscript PDF file)?

Reviewer #2: No

Reviewer #4: (No Response)

5. Is the manuscript presented in an intelligible fashion and written in standard English?

Reviewer #2: Yes

Reviewer #4: Yes

6. Review Comments to the Author

Reviewer #2: The authors have addressed reviewers comments satisfactorily

Reviewer #4: Reviewer Comments: RE: Singh et al, melioidosis risk factors

Reviewing revised manuscript with tracked changes.

Major:

I think the article reads well after the first round of revisions. I do not have any major comments!

Flow diagram in appendix is good.

Minor:

1. Line 10: change title to “January 2017-July 2018” (rather than August 2018).

2. Line 52: “in the tropics”

3. Line 62: Tsunamis

4. Line 66-67: Why is this the case? Is it because laboratory diagnosis is limited in other centres?

5. Line 77-78: Suggest rewording along lines of “identify risk factors to inform the initial public health response”.

6. Line 86: “automated system” – which system? BacT/ALERT as per CSF or different?

7. Line 91: I think type 3 secretion system should be abbreviated: TTSS or T3SS not TTSI as brackets.

8. Line 186-187: How significant was this association? Worth putting in the text?

9. Line 190: Suggest removing the word bacteria and just having “aerosolized B. pseudomallei”

10. Line 218: “We recommend sensitizing medical doctors..”- while I understand what you mean by this, it reads like a medical procedure! Would suggest something along the lines of “We recommend increased education of medical..”

11. Table 1, line 370: Multiple chronic comorbidities is a listed risk factor, however is this 2 or more comorbidities? How was this defined? I think this should be defined better including potentially listing (as an endnote) what you called a comorbidity.

7. PLOS authors have the option to publish the peer review history of their article (what does this mean?). If published, this will include your full peer review and any attached files.

**Do you want your identity to be public for this peer review?** For information about this choice, including consent withdrawal, please see our Privacy Policy.

Reviewer #2: **Yes: **Kavitha Saravu

Reviewer #4: No

---

## [Decision Letter · Decision Letter 2]

15 Nov 2022

Risk Factors for Melioidosis in Udupi District, Karnataka, India, January 2017-July 2018

PGPH-D-22-00623R2

Dear Dr Dikid,

We are pleased to inform you that your manuscript 'Risk Factors for Melioidosis in Udupi District, Karnataka, India, January 2017-July 2018' has been provisionally accepted for publication in PLOS Global Public Health.

Best regards,

Raquel Muñiz-Salazar, Ph.D.

Academic Editor

After two revision processes, the authors have addressed all comments.

The decision is to ACCEPT it.

Reviewer Comments (if any, and for reference):

Reviewer's Responses to Questions

**Comments to the Author**

1. If the authors have adequately addressed your comments raised in a previous round of review and you feel that this manuscript is now acceptable for publication, you may indicate that here to bypass the “Comments to the Author” section, enter your conflict of interest statement in the “Confidential to Editor” section, and submit your "Accept" recommendation.

Reviewer #4: All comments have been addressed

2. Does this manuscript meet PLOS Global Public Health’s publication criteria? Is the manuscript technically sound, and do the data support the conclusions? The manuscript must describe methodologically and ethically rigorous research with conclusions that are appropriately drawn based on the data presented.

Reviewer #4: Yes

3. Has the statistical analysis been performed appropriately and rigorously?

Reviewer #4: Yes

4. Have the authors made all data underlying the findings in their manuscript fully available (please refer to the Data Availability Statement at the start of the manuscript PDF file)?

Reviewer #4: (No Response)

5. Is the manuscript presented in an intelligible fashion and written in standard English?

Reviewer #4: Yes

6. Review Comments to the Author

Reviewer #4: Happy that all of my comments addressed.

7. PLOS authors have the option to publish the peer review history of their article (what does this mean?). If published, this will include your full peer review and any attached files.

**Do you want your identity to be public for this peer review?** For information about this choice, including consent withdrawal, please see our Privacy Policy.

Reviewer #4: No
